# Potential Distribution of Goldenrod (*Solidago altissima* L.) during Climate Change in South Korea

**Jeong Soo Park \*, Donghui Choi and Youngha Kim**

Division of Ecological Safety, National Institute of Ecology, Seocheon 33657, Korea; dhchoi82@nie.re.kr (D.C.); khatru@nie.re.kr (Y.K.)

\* Correspondence: jspark@nie.re.kr; Tel.: +82-10-7999-9705

**Abstract:** Predictions of suitable habitat areas within a specific region can provide important information to assist in the management of invasive plants. Here, we predict the current and future potential distribution of *Solidago altissima* (tall goldenrod) in South Korea using climatic and topographic variables and anthropogenic activities. We adopt four single models (the generalized linear model, generalized additive model, random forest, and an artificial neural network) and a weighted ensemble model for the projection based on 515 field survey points. The results showed that suitable areas for *S. altissima* were mainly concentrated in the southwest regions of South Korea, where temperatures are higher than in other regions, especially in the winter season. Solar radiation and Topographic Wetness Index (TWI) were also positively associated with the occurrence of *S. altissima*. Anthropogenic effects and distances from rivers were found to be relatively less important variables. Based on six selected explanatory variables, suitable habitat areas for *S. altissima* have expanded remarkably with climate changes. This range expansion is likely to be stronger northward in west coastal areas. For the SSP585 scenario, our model predicted that suitable habitat areas increased from 16,255 km$^2$ (16.2% of South Korea) to 44,551 km$^2$ (44.4%) approximately over the past thirty years. Our results show that *S. altissima* is highly likely to expand into non-forest areas such as roadsides, waterfront areas, and abandoned urban areas. We propose that, based on our projection maps, *S. altissima* should be removed from its current margin areas first rather than from old central population areas.

**Keywords:** invasive plant; *Solidago altissima* L.; ensemble model; suitable habitat; climate change

## 1. Introduction

The spread of invasive plant species into native vegetation has reduced biodiversity and altered landscape structures and ecosystem functions while also having harmful effects on the social economy and human well-being [1–3]. The establishment and expansion of invasive species depends on their biological traits, environmental conditions, and competition with native species [4]. Because successful invasive plants have a broad environmental tolerance range, superior competitiveness, and dispersion ability compared to other species, invasive plants often extend to new habitats rapidly, and the rate of their spread does not decline in most cases [5,6].

*Solidago altissima*, a goldenrod species, is among the exceptionally successful invaders in Europe, Australia, New Zealand, Japan, and South Korea [7–9]. Owing to its high abundance and rapid extension into habitats of native vegetation, several countries designated this species as a harmful invasive species and have made efforts to control its population [10]. The native range of *S. altissima* covers large parts of North America, from Florida in the USA to Ontario in Canada [11]. After this

species was introduced into Europe in the 17th or 18th century, it expanded to most of the European continent, from Scandinavia to northern Italy [12]. In the case of South Korea, populations of *S. altissima* have frequently been found in southwest areas since 1970 [10].

Maps of current and future suitable habitat areas can provide important information pertaining to the management of invasive plants such as their establishment and growth, climatic limitations, and potential distribution ranges. Specifically, more funding and effort can be allocated to climatically suitable areas to control the spread of these species [13]. Several researchers have emphasized that invasive species should be removed at their current marginal areas first, in accordance with the potential distribution maps [14,15]. Additionally, rising temperatures and changing rainfall patterns stemming from climate change will likely affect the distribution of invasive plants [16]. Projection maps can show the spread direction and speed of such plants under climate change scenarios.

The aim of this study is to answer several questions related to the distribution of *S. altissima*: Which environmental factors are significantly related to the occurrence of *S. altissima*? Where are suitable habitats for this invasive species in South Korea based on current climate conditions? How much will suitable habitat areas change under climate change?

## 2. Materials and Methods

### 2.1. Study Area

The study area was limited to the southern part of the Korean Peninsula (South Korea) that extends from 33°0′ to 38°9′ N and 124°5′ to 132°0′ E. Almost 70% of the surface area of South Korea (total = 100,340 km$^2$) is mountainous, with mountain ranges mainly situated in the east (Figure 1). South Korea has a high level of vascular plant species richness (4552 taxa) due to heterogeneity in its topography and climate [17]. The main climate types in South Korea are monsoon-influenced hot-summer humid continental climate in the northern inland areas and a humid subtropical climate in the southern coastal area according to the Köppen climate classification. The mean annual temperature is 12.5 °C, and the mean annual precipitation is approximately 1300 mm over the last 30 years [18]. The Korean Peninsula has low levels of winter and spring precipitation, and two-thirds of the annual precipitation occurs in the summer (i.e., the monsoon season). Habitat types were classified into eight categories (i.e., forest, forest edge, grassland, roadside, residential area, farmland, waterfront area, coast) with the presence of *S. altissima*.

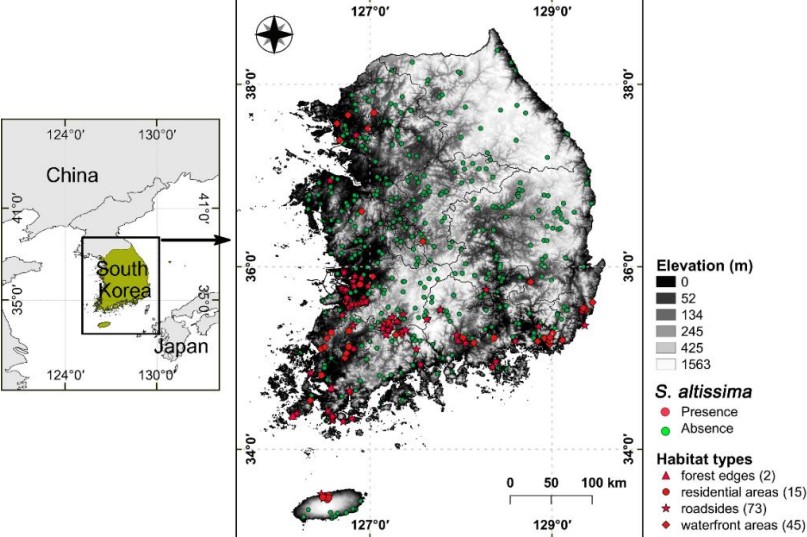

**Figure 1.** Map of field survey plots and elevation range of South Korea. The red and green dots indicate a total of 515 data points (red dots show the presence of *Solidago altissima* (*n* = 135) and green dots show areas without *S. altissima* (*n* = 380)).

## 2.2. Study Species

*Solidago altissima* L. (late goldenrod) was introduced into South Korea as a nectar and ornamental plant before 1970 [19]. Since then, this species has expanded rapidly into the southern part of the Korean Peninsula. The Ministry of the Environment of the Republic of Korea designated this species as a harmful invasive alien species in 2008 due to its substantial environmental impact [19]. *S. altissima* from North America is also considered to be an exceptionally successful invader in countries such as Europe, Japan, China, Australia, and New Zealand. *S. altissima* can rapidly become dominant and remain for long durations in nutrient-rich and stable, moist soil conditions [8]. Furthermore, because this species builds up dense stands with a large below-ground biomass, it has superior competitive ability compared to native plants [2]. This plant can reproduce in its first year, and individual shoots produce more than 20,000 seeds [20]. Whereas seeds can allow long-distance dispersal due to their small size, colonial extension of an established population mainly occurs via rhizomes [8].

## 2.3. Model Variables

Distribution data for *S. altissima* were mainly obtained from a national survey of non-native species in Korea conducted from 2015 to 2019. We collected 135 presence data points and selected randomly 380 absence data points throughout South Korea. We took into account a total 27 environmental factors for our model (Table S1): temperature (bio1–bio11), precipitation (bio12–bio19), solar radiation, topography variables (i.e., elevation, aspects, slopes, Topographic Position Index (TPI), Topographic Wetness Index (TWI), and distances from rivers), and the global human foot-printing (HFP) dataset from the SocioEconomic Data and Applications Center (SEDAC, http://sedac.ciesin.columbia.edu/) were also included in our modeling [21]. We extracted elevation, aspect, slope, TPI, and TWI data from the Digital Elevation Model (DEM, 1 km × 1 km resolution). Potential incoming solar radiation and TWI data were implemented with the SAGA-GIS modules (www.sagsgis.org) in QGIS [22]. The aspect was divided into the following four categories: 1 = 315°–45°, 2 = 45°–90° or 270°–315°, 3 = 90°–135° or 225°–270°, and 4 = 135°–225°. Positive TPI values represent locations that are higher than the average of their surroundings. Negative TPI values represent locations that are lower than their surroundings. TWI values were calculated to estimate the amount of moisture in the soil. It is defined as $\ln(a/\tan\beta)$, where a is the local upslope area, and $\tan\beta$ is the local slope [23]. Current (1970–2000) and future (2021–2040 and 2041–2060) climate datasets of the Coupled Model Intercomparison Project Phase 6 (CMIP6) were downloaded from WorldClim version 2.1 [24]. We selected two Shared Socio-economic Pathways (SSP245, SSP585) based on the Global Circulation Model (GCM) managed at the Beijing Climate Center (BCC-CSM2-MR). All raster explanatory variables used identical spatial extent, resolutions (1 km × 1 km), and geographic coordinate systems (WGS84, EPGS 4326), with the help of the bilinear method in R. To avoid collinearity, which can lead to incorrect estimations, we moved non-independent variables relative to others based on the pairwise Pearson correlation coefficient (r pairwise ≥ 0.7) (Figure S1). Next, multicollinearity was tested with the variance inflation factor (VIF ≥ 3) [25]. Finally, we selected nine explanatory variables considering the eco-physiological characteristics of *S. altissima* (i.e., annual mean air temperature (bio1), temperature annual range (bio7), annual precipitation (bio12), precipitation during the driest quarter (bio17), solar radiation, TWI, TPI, HFP, and distances from rivers.

## 2.4. Species Distribution Modeling

To model the suitability of *S. altissima* in South Korea, we included four single-model algorithms available in the Biomod2 library: the generalized linear model (GLM), the generalized additive model (GAM), random forest (RF), and an artificial neural network (ANN). The models were processed according to the default settings of Biomod2 [26]. Although these modeling methods are commonly used and show high performance capabilities in cases of species distribution modeling, we applied ensemble approaches to decease the predictive bias of the single models by combining their projections [27]. We used an AUC-weighted combining method, which is known to predict species distributions fairly

well [28–30]. Initially, the AUC values of the single models were calculated for the single-model projections. Subsequently, the output layers of the single-model projections were combined in an ensemble forecast layer [31]. The ensemble approach was implemented with the raster calculator in QGIS. We hypothesized that *S. altissima* cannot spread into the forest areas, according to previous study [8]. Forest areas were overlaid on potential distribution maps and removed from the result maps. Forest areas were calculated based on the land cover map of the Ministry of the Environment [32].

The extracted data points (*n* = 515) were randomly divided into the model training dataset and the model evaluation dataset at a ratio of 7:3. During the data splitting process, the ratio between the number of presences and absences in the training and evaluation processes was kept constant [33].

Single-model and ensemble-model accuracy levels were determined by the AUC values of the receiver operating characteristic (ROC) curves, kappa statistics, the sensitivity level (omission error), and by specificity (commission error) [34,35]. To convert continuous model predictions to a binary classification, we used a threshold value that maximized the sum of the sensitivity and specificity outcomes [36]. We considered AUC values below 0.7 as poor, those in the range of 0.7–0.9 as moderate, and those above 0.9 as good [37]. Kappa ranged from −1 to +1, where +1 indicates a perfect match between the observation and the prediction, and a value below 0 indicates an outcome no better than a random classification [38]. We used the following ranges to interpret Kappa statistics: values of <0.4 were poor, 0.4–0.8 useful, and >0.8 good [39]. The AUC values and Kappa statistics were calculated with the PresenceAbsence package for R 3.6.3 [40].

We applied GAMs to estimate the relationships between the occurrences of *S. altissima* and the selected explanatory variables in R (version 3.6.3; R Development Core Team, 2020). A binomial distribution was specified with a logit link function. The explanatory variables were modeled as cubic splines, with six degrees of freedom for smoothing splines [41].

## 3. Results

### 3.1. Variable Importance

We evaluated the explanatory variables' importance and calculated the ranges of each variable with respect to the probability of the occurrence of the species. The response curves of GAM showed that the probability of *S. altissima* presence increased as bio1, bio12, TWI, and solar radiation increased, and it decreased as bio7 increased (Figure 2). The response curves for bio7 revealed a sharp decrease in the predicted presence of the species above an annual temperature range of 35 °C. The presence probabilities of *S. altissima* for annual precipitation (bio12) increased with values of up to 1600 mm and then did not increase. Based on the mean decrease accuracy and deviance explained in GAM, the annual mean air temperature (bio1), temperature annual range (bio7), and precipitation of the driest quarter (bio17) were decisive factors determining the distribution of *S. altissima*. In contrast, TPI, HFP, and distances from rivers did not significantly affect the occurrence of this plant compared to other variables.

All models showed good performance based on the AUC value (>0.70) and kappa (>0.40). Compared with other models, the random forest model yielded the highest performance, while GLM had the lowest predictive power (Table 1).

**Table 1.** Model validation statistics and threshold (range: 0–100) for suitable habitats of *S. altissima.*

| Models | | Sensitivity | Specificity | AUC | Kappa | Threshold |
|---|---|---|---|---|---|---|
| Single model | GLM | 0.70 | 0.77 | 0.80 | 0.42 | 56 |
| | GAM | 0.89 | 0.81 | 0.90 | 0.61 | 57 |
| | RF | 0.92 | 0.95 | 0.98 | 0.85 | 33 |
| | ANN | 0.83 | 0.81 | 0.87 | 0.56 | 62 |
| Ensemble Model | | 0.85 | 0.87 | 0.95 | 0.67 | 49 |

Abbreviation: AUC = area under curve, GLM = generalized linear model, GAM = generalized additive model, RF = random forest, ANN = artificial neural network, EM = ensemble model.

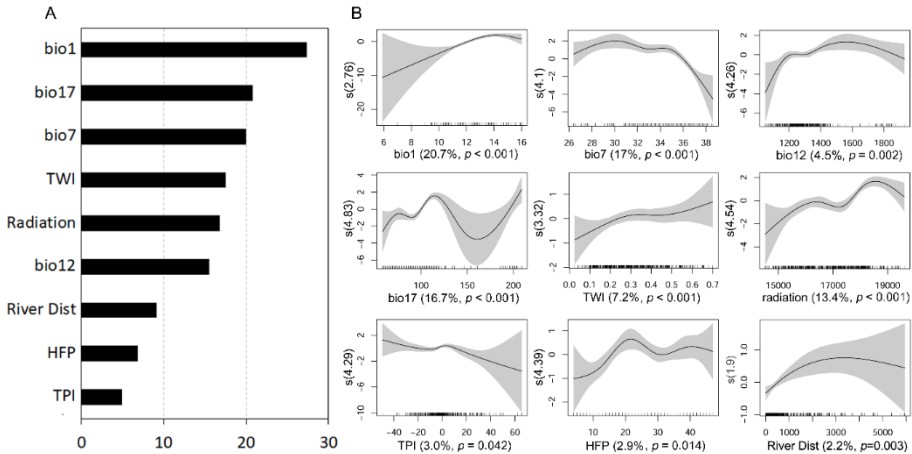

**Figure 2.** Mean decrease accuracy of explanatory variables as assigned by random forest (**A**). Generalized additive model (GAM) response curves depicting the relationship between the probabilities of the presence of *S. altissima* on a logit scale and nine explanatory variables (**B**). The shaded bands indicate the 95% confidence interval. The degree of smoothing is indicated by the *Y*-axis brackets. Deviance explained (%) values and the significance of the explanatory variable (*p*-value) are depicted by the *X*-axis brackets.

### 3.2. Current Habitat Suitability

When we estimated the current suitable habitat of *S. altissima* based on the threshold values (maximum sensitivity plus specificity), the suitable habitats, except for the forest area, were found to account for approximately 9.3–13.0% of the surface area of South Korea according to the five models (Figure 3). Maps showed that the suitable areas for *S. altissima* were mainly concentrated in the southwest region of South Korea. For the weighted ensemble model, we estimated that suitable habitat areas were characterized by elevations of 0–715 m in comparison with unsuitable areas at 0-1839 m, with an annual mean air temperature of 10–16 °C in comparison with unsuitable areas at 4.7–16.1 °C, and with an annual mean precipitation of 1076–1919 mm in comparison with unsuitable areas at 1048–2209 mm.

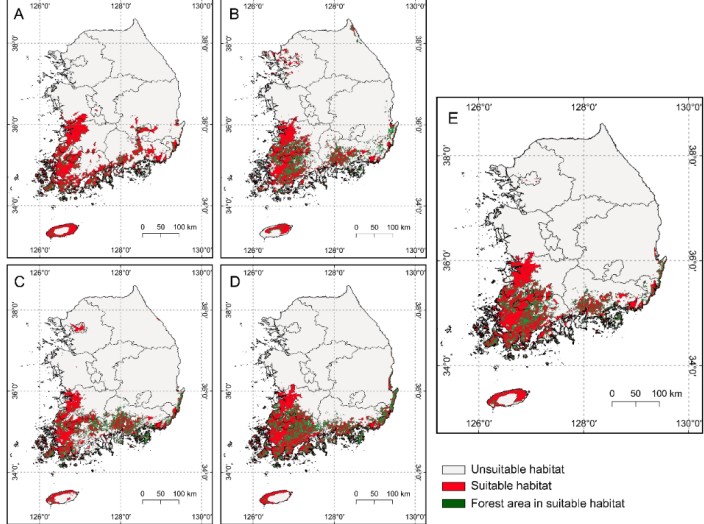

**Figure 3.** Maps of the current suitable habitat (red zone) for *S. altissima* (**A**): generalized linear model (the range of the red zone except for the forest area was 13.0% of South Korea), (**B**): generalized additive model (9.3%), (**C**): random forest (9.6%), (**D**): artificial neural network (11.8%), (**E**): weighted ensemble model (11.1%)).

### 3.3. Future Habitat Suitability under Climate Change Scenarios

The suitable habitats changes for *S. altissima* were predicted by the weighted ensemble model. The results showed that the areas of suitable habitat (habitat suitability index > 50) increased gradually for the periods of 2021–2040 and 2041–2060 under the climate change scenarios (Figure 4). The suitable habitat areas were concentrated in the west and south coastal regions clearly and extended toward the northern areas with climate change.

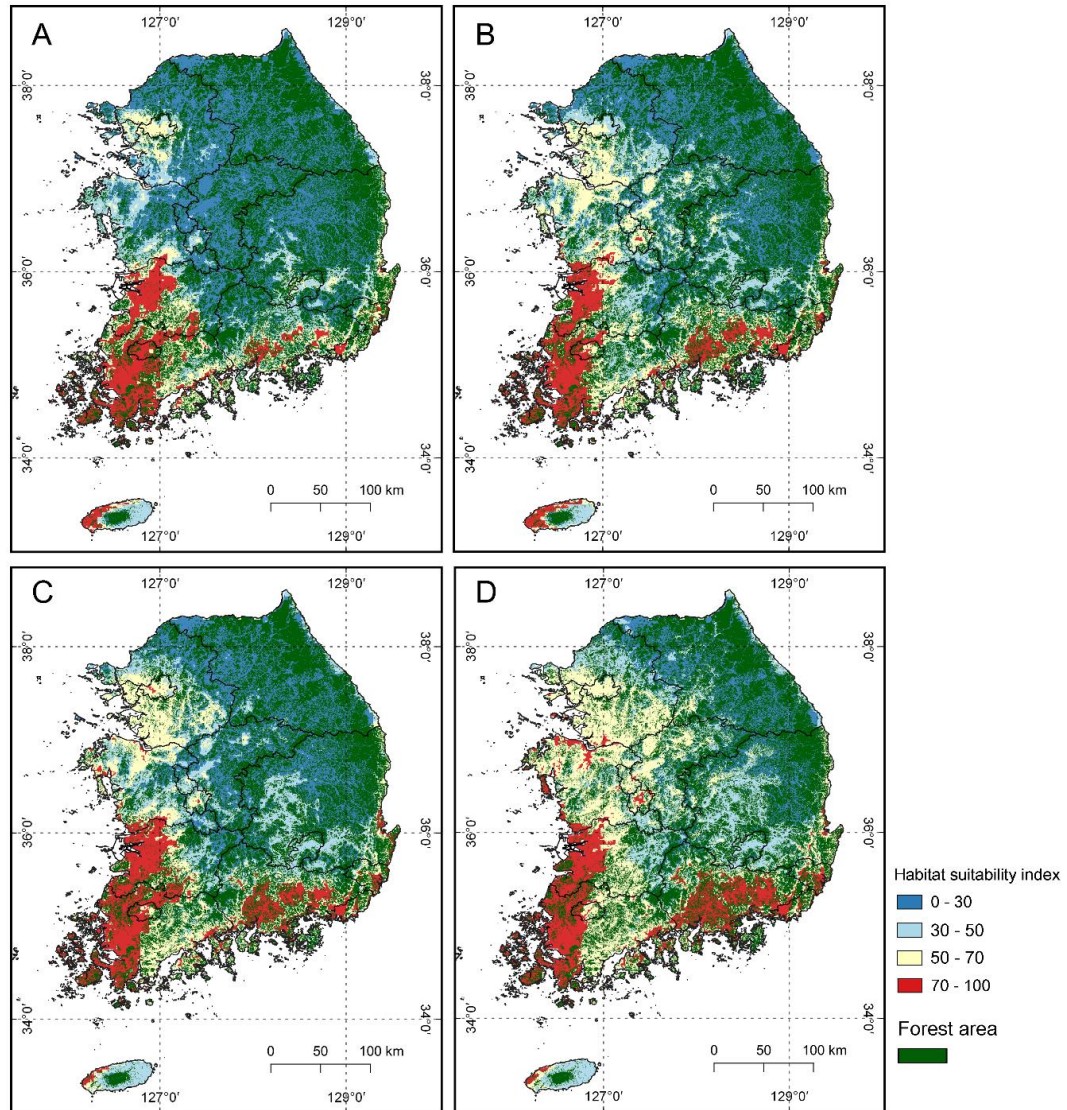

**Figure 4.** Maps of the suitable habitat changes for *S. altissima* under two climate changes scenarios (SSP245 and SSP585) for the periods of 2021–2040 and 2041–2016 taken from the global model at a resolution of 1 km ((**A**): SSP245, 2021–2040, (**B**): SSP585, 2021–2040, (**C**): SSP245, 2041–2060, (**D**): SSP585, 2041–2060).

Using the four grades of habitat suitability classification, we found the suitable habitat areas which excluded the forest areas (habitat suitability index > 50) increased from 17,107 km$^2$ (17% of total South Korea) to 23,417 km$^2$ (23.3%) for the periods of 2021–2040 and 2041–2060 under scenario SSP245 (Figure 5). For the SSP585 scenario, our model predicted that the suitable habitat area overall increased from 21,972 km$^2$ (21.9% of total South Korea) to 31,193 km$^2$ (31.1%) over approximately twenty years.

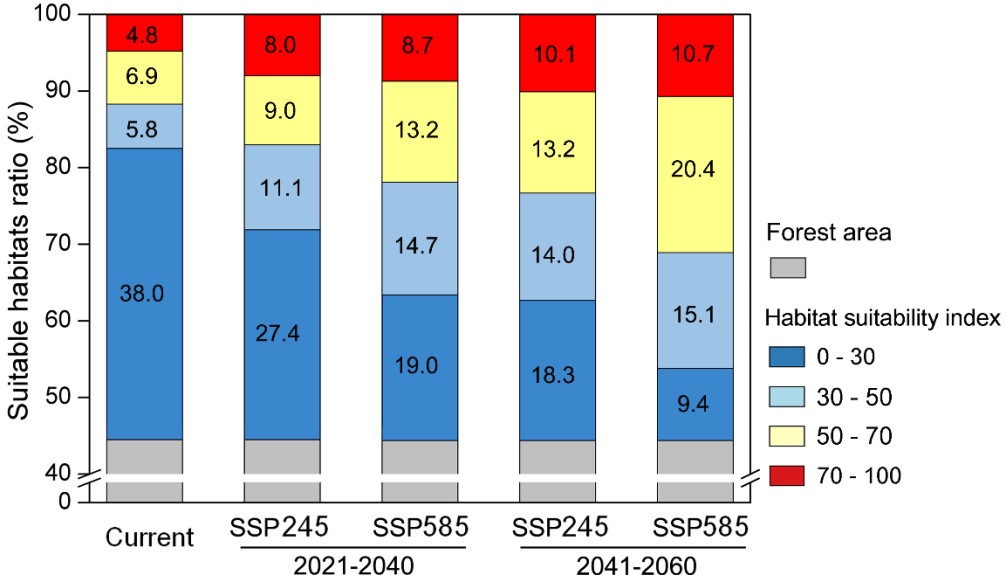

**Figure 5.** Change of suitable habitat ratios for *S. altissima* under two climate change scenarios. Forest areas were excluded from total suitable habitat map.

## 4. Discussion

This study estimated the associations between the distributions of *S. altissima* and environmental variables. GAM response curves showed that the occurrence of *S. altissima* was positively associated with the annual mean air temperature (from 6 to 14 °C) A previous study also estimated that *S. altissima* is very likely to expand southward rather than northward in Europe [13]. In the case of the Korean Peninsula, Bio1 derived from the WorldClim dataset showed a strong correlation (−0.68) with elevation (Figure S1), which indicated that *S. altissima* is mainly distributed in non-forested areas located at low elevations; the altitude range of the present plots of *S. altissima* were from 0 to 437 m a.s.l. (Table S1). Precipitation in the driest quarter (bio 17) and the annual temperature range (bio 7) were also significantly associated with the distribution of this species in South Korea, indicating that *S. altissima* prefers areas with high winter precipitation levels and mild winter temperatures (Figure S2). The Korean Peninsula has little precipitation in winter and early spring. Additionally, the inland areas have a higher annual temperature range than the southern coastal area [18]. Despite the fact that *S. altissima* has large native ranges in northern America (from southern Florida in the US to Canada along the east coast) [42], dry and cold winter weather may function as a stress factor and prevent the establishment of this plant.

Solar radiation and TWI were positively associated with the occurrence of *S. altissima* (Figure S2). Earlier work reported that *S. altissima* is a light-demanding species, and it cannot reproduce under shaded conditions [8]. For this reason, this plant cannot invade dense forested areas. Furthermore, the forests of South Korea consist of three types of secondary succession forests: pine forests, pine-oak forests, and oak mixed forests [43]. We can speculate that the allelopathy substance of pine litter and the shading of oak tree canopies may make it impossible for *S. altissima* to expand into forests. This species is generally found in prairies, along roadsides, at riverbanks, and along forest edges in the US and Europe [42], similar to the distribution patterns in South Korea. Moreover, Weber (2000) reported that *S. altissima* prefers moist soil [8], and this species shows major variations in biomass depending on the soil nutrients [44]. Generally, the soil of estuaries has more nutrients and shows higher TWI scores compared to the soils of other regions. A field survey also showed that the population size and density were higher in estuary regions than in upstream areas in the YoungSan river of South Korea [45]. In Germany, *S. altissima* became dominant sooner in nutrient-rich clay-containing soil than in nutrient-poor sandy soil [46]. It is well known that among goldenrod species, *S. gigantea* and

*S. altissima* are now the most serious invaders across Europe following their introduction in the 17th or 18th century [8,47,48]. In terms of the soil moisture regime, *S. gigantea* is frequently found on moist soil with relatively stable moisture levels over time, while *S. altissima* tends to be found densely growing on intermediate-moisture and well-drained soils [47]. We can speculate that *S. altissima* can invade rapidly into waterfront areas due to appropriate environmental conditions such as rich nutrients, stable soil moisture, and sufficient solar radiation.

Our results show that distances from rivers and anthropogenic effects were relatively less important variables compared to other variables. When we classified the habitat types of the presence areas of *S. altissima* in South Korea (n = 135), roadsides (54%) had the greatest proportion, followed by waterfront areas (33%). Residential areas and forest edges accounted for only 11% and 2%, respectively. While the distribution of *S. altissima* was positively associated with TWI on the landscape scale, this species is more frequently found at roadsides rather than waterfront areas on a local scale. As mentioned above, *S. altissima* prefers intermediate moisture soil and can tolerate a wide range of soil-moisture conditions compared to other goldenrod species [47]. For this reason, this species can expand along roadsides as well as waterfront areas [13]. Our results showed that these anthropogenic effects (human foot-printing) were not a critical factor with regard to dispersal. The seeds *of S. altissima* are adapted for long-distance dispersal by wind, and planting is prohibited by law because this species was designed as a harmful invasive alien species by the Ministry of the Environment.

Based on the six selected explanatory variables, we predicted current and future suitable habitat areas under climate change scenarios. Our models showed excellent accuracy levels. In other words, the range of the current suitable habitat area is fairly similar to the observed range in South Korea for this species, which can be attributed to its relatively limited geographical range, the strong associations with climate factors, and the minor effects of anthropogenic factors on the dispersal of this species [49,50]. In terms of climate, *S. altissima* may be prevalent in areas with higher annual temperatures, mild winter temperatures, and higher precipitation levels during the driest season. Indeed, suitable habitats are concentrated in the southwest parts of the Korean Peninsula, and field survey results have shown that the population sizes of *S. altissima* are larger in southern areas than in northern areas [45].

Suitable habitat areas for *S. altissima* have expanded remarkably along with climate change. Range expansion is likely to be stronger northward in western coastal areas. Temperature and precipitation at the study sites (n = 515) increased steadily under the climate change scenarios here, showing an annual mean air temperature increase of 2.6 °C and an annual precipitation increase of 93 mm according to the SSP 585 climate change scenario over the next 30 years [24]. These results reveal that rising temperatures and increasing precipitation levels can exert a positive effect on the expansion of this species. Weber (2001) estimated that *S. altissima* has the largest potential range in its latitudinal extent among the three goldenrod species introduced into Europe [13].

Although our modeling results can provide a broad basis for invasive species management on a national scale, projections of current and future distributions need to be interpreted with caution due to several limitations here. We did not account for biotic interactions or soil physicochemical properties on a local scale. For example, the shading of tree species and competition with other herbaceous species can be critical limitations with regard to how this species can expand. At the early stage of invasion, nutrient-rich soil and moderate soil water content levels can facilitate the establishment and rapid expansion of this species. Future models should incorporate a wide variety of factors that interact with species-specific physiological characteristics [51].

## 5. Conclusions

Our results have demonstrated that *S. altissima* is highly capable of expanding into non-forested areas such as roadsides, waterfront areas, and abandoned urban areas. The growth of *S. altissima* can be controlled by a combination of mowing and soil rotation [13]. Subsequently, the sowing of tall grass species can suppress the regeneration of this invasive species and prevent the soil from being washed

away. Next, based on our projection maps, invasive plant managers should prioritize eradicating *S. altissima* in margin areas of its current range rather than in areas of dense populations.

**Supplementary Materials:** The following are available online at http://www.mdpi.com/2071-1050/12/17/6710/s1, Table S1: Comparison of mean and range of environmental variables between the presence and absence spots of *S. altissima* in South Korea, Figure S1: Pearson correlation coefficient matrix comparing 27 environmental variables, Figure S2: Maps of six environmental variable patterns in South Korea.

**Author Contributions:** J.S.P. analyzed the data and wrote the paper. D.C. and Y.K. provided technical assistance to J.S.P. and helped with field data collection. All authors have read and agreed to the published version of the manuscript.

**Funding:** This research was supported by National Institute of Ecology (No. NIE-A-2020-08).

**Acknowledgments:** We would like to thank the Ministry of Environment of the Republic of Korea for its assistance. We are also grateful to all the researchers who carried out field surveys.

**Conflicts of Interest:** The authors declare no conflict of interest.

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
