# Peer review of "Potential Distribution of Goldenrod (Solidago altissima L.) during Climate Change in South Korea"

_sustainability, doi:10.3390/su12176710_

Round 1
Reviewer 1 Report
Dear Authors!
Although overall technically correctly written, I have found in your manuscript couple of things that I think should be corrected prior to its publishing.
First, decide how many explanatory variables you have been used for your models. In the abstract (Line 18) and the Discussion (Line 266) you state that models are based on six explanatory variables, but in M&M (Lines 106-109) and in the Results (Lines 143-153 and Figure 2) there are nine explanatory variables!
It will be usefully for potential readers of your manuscript if you explain in little more details what are the TWI and TPI! The former you are mentioning in the Abstract with full name, but it should be written in M&M section as well. How have you used data on the aspect? As nominal variables, degree values or some transformation of degree value?
Figure 4 – there is typo error in Line 198. It should be “2040” instead of “2014”
On several occasions, you are writing “S. gigantean”. As far as I am aware, it is “S. gigantea”, without “n” at the end.
In general, what I am missing most in your manuscript is the fact that you have not overlay your models with some map of habitats/vegetation of South Korea. After that, you will get more realistic results in terms of goldenrod potential distribution. What is the purpose of predicting distribution of the goldenrod on parts that are covered by forest vegetation, if goldenrod cannot spread into such habitats?
I think you should completely rewrite your conclusion with stronger connection to your presented results. First sentence of your conclusion is not supported by presented methods and results, while second has nothing to do with your research but is a citation.
Author Response
Response to Reviewer 1 comments
Point 1: How many explanatory variables have you been used for the models?
Response 1: first of all, we used 27 explanatory variables (Table S1). To avoid collinearity, we calculated Pearson correlation coefficient to removed non-independent variables (Figure S1) and selected 9 variables. Finally, we selected 6 explanatory variables based on variable importance and GAM results (Figure 2). (line 93-98)
Point 2: Explain the meaning of TWI and TPI in detail.
Response 2: We mentioned the meaning of TPI and TWI in detail (line 102-105)
Point 3: How have you used aspect data?
Response 3: The aspect was divided into the following four categories, and described it in detail (line 100-102)
Point 4: Overlay model results with habitats/vegetation map to get more realistic result.
Response 4: We overlaid the forest area maps with our model results, and removed the forest areas from the suitable habitats maps (Figure 3, 4, 5)
Point 5: Rewrite conclusion with strong connection to results
Response 5: We revised the discussion to get strong connection with results (line 223-225), (line 229-231), (line 255-262). We removed the sentences which have weak connections with results. (line 247-253 in original manuscript)
Lastly, we corrected the typo and punctuation error according to your comments. Grammars and spelling of manuscript checked by a native English speaker.

Reviewer 2 Report
The article is well written and shows a well-designed statistical model for making reliable predictions about the alien species considered. The subject matter is highly topical and pertinent to the issues of the journal. Therefore in the attached file I have highlighted only some typos, some suggestions for some relevant citations and the request for a better explanation of acronyms. After a minor revision the paper will be ready for the publication in "Sustainability"
Author Response
Response to Reviewer 2 comments
Point 1: Quote additional references.
Response 1: We quoted two previous studies (line 335-336), (line 421-423)
Point 2: Explain environmental variables clearly
Response 2: We explained total explanatory variables in detail. (Table S1), (line 93-98)
Point 3: Explain the meaning of TWI and TPI in detail.
Response 3: We mentioned the meaning of TPI and TWI in detail (line 102-105)
Lastly, we corrected the typo and punctuation error according to your comments. Grammars and spelling of manuscript checked by a native English speaker.
